# Graph Anomaly Detection with Bi-level Optimization

## ABSTRACT

Graph anomaly detection (GAD) has various applications in finance, healthcare, and security. Graph Neural Networks (GNNs) are now the primary method for GAD, treating it as a task of semi-supervised node classification (normal vs. anomalous). However, most traditional GNNs aggregate and average embeddings from all neighbors, without considering their labels, which can hinder detecting actual anomalies. To address this issue, previous methods try to selectively aggregate neighbors. However, the same selection strategy is applied regardless of normal and anomalous classes, which does not fully solve this issue. This study discovers that nodes with different classes yet similar neighbor label distributions (NLD) tend to have opposing loss curves, which we term it as "loss rivalry". By introducing Contextual Stochastic Block Model (CSBM) and defining *NLD distance*, we explain this phenomenon theoretically and propose a **Bi**-level **o**ptimization **G**raph **N**eural **N**etwork (BioGNN), based on these observations. In a nutshell, the lower level of BioGNN segregates nodes based on their classes and NLD, while the upper level trains the anomaly detector using separation outcomes. Our experiments demonstrate that BioGNN outperforms state-of-the-art methods and effectively mitigates "loss rivalry". Codes are available at https://anonymous.4open.science/r/BioGNN-12B4.

## CCS CONCEPTS

• **Security and privacy** → **Web application security**; • **Computing methodologies** → **Neural networks**.

## KEYWORDS

Graph Neural Networks, Anomaly Detection, Bi-level Optimization

## 1 INTRODUCTION

Graph anomaly detection (GAD) is a learning-to-detect task. The objective is to differentiate anomalies from normal ones, assuming that the anomalies are generated from a distinct distribution that diverges from the normal nodes [22]. As demonstrated by [29], GAD has various real-world applications including detecting spam reviews in user-rating-product graphs [18], finding misinformation and fake news in social networks [10], and identifying fraud in financial transaction graphs [30, 40].

A primary method is to consider GAD as a *semi-supervised node classification* problem, where the edges play a crucial role. By examining the edges, we can divide an ego node's neighbors into two groups: (1) homophilous neighbors that have the same labels as

**Figure 1: The ego normal node and anomaly (marked in red circle) have comparable neighborhood label distributions (NLD). The probability of neighbor labels being 0 or 1 is denoted by $p_c$ and $q_c$, where the subscript represents the class label. For instance, $p_1$ denotes the probability of a normal neighbor for anomalies.**

the ego node, and (2) heterophilous neighbors whose labels are different from the ego node's label. For instance, in the case of an anomaly ego node, its interactions with anomaly neighbors display homophily, while its anomaly-normal edges demonstrate heterophily. Both homophily and heterophily are prevalent in nature. In transaction networks, fraudsters have heterophilous connections with their customers, while their connections with accomplices are homophilous.

From the standpoint of neighbor relationships, we can briefly describe the primary graph neural networks (GNNs)-based GAD solutions and their limitations as follows:

- Early studies [26, 41] aggregate over all neighbors without considering the impact of homophily and heterophily. That is, the representation of each node blindly aggregate the information from all neighbors, without discriminating the neighbor relationships. However, this approach can be disadvantageous to GAD as anomalies are more likely to be hidden among a large number of normal neighbors. Blindly aggregating information can dilute the suspiciousness of anomalies with normal signals, making them less discernible [18, 25, 30, 37].
- To address the above-mentioned problem, recent studies [3–5, 11, 17, 40] draw inspiration from graph signal processing (GSP). They suggest that a low-pass filter may not be optimal for all graphs. Instead, they manipulate eigenvalues of the normalized graph Laplacian to amplify some frequency information and weaken others. However, these studies optimize node representations as a whole, without addressing differences in their distribution regarding neighbor labels. For instance, as shown in Figure 1, a normal node shares the same neighbors as an anomaly. Our analysis in §3.2 reveals that nodes of different classes with the same neighbors retain rather different frequency components. While emphasizing a single frequency band can improve learning for some nodes, it can hinder the learning of others.

Thus, it is crucial to understand the impact of neighbor label distribution (NLD) on detector behavior. We introduce and reveal the phenomenon of "loss rivalry". Surprisingly, we observe opposite loss curves for anomalies and normal nodes holding similar NLDs.

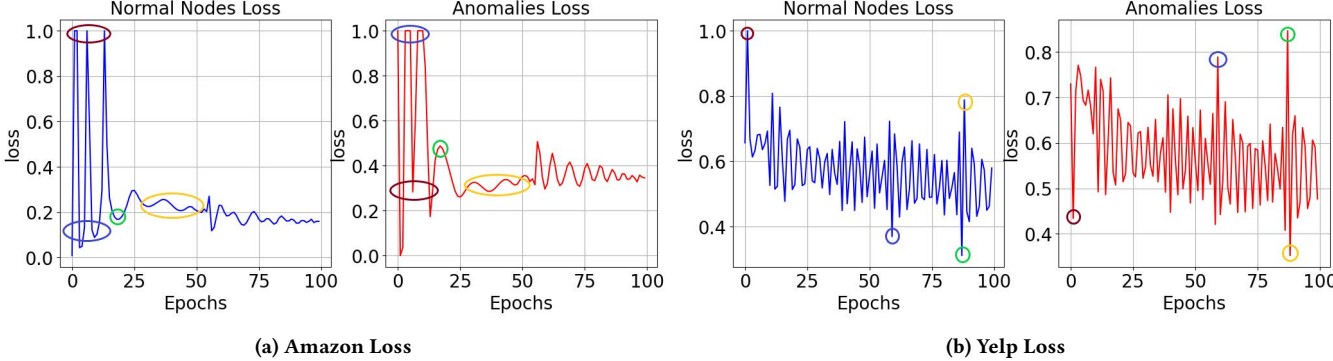

(a) Amazon Loss

(b) Yelp Loss

Figure 2: Illustration of the 'loss rivalry' phenomenon in YelpChi and Amazon Datasets with BWGNN [40]. From the same-color circles around the maxima and minima, we observe that the two loss curves in the same dataset are opposite along the epochs. The curves are plotted

These are separately highlighted around the maxima and minima of the curves in Figure 2. Our analysis emphasizes the importance of using distinct aggregation mechanisms for nodes with different classes but similar NLD.

Based on this finding, we propose a bi-level optimization model in §4, named BioGNN. Specifically, it consists of two key components. The first component is a mask generator that separates nodes into mutually exclusive sets based on their classes and NLD. The second component contains two well-designed GNN encoders that adopt different mechanisms to learn node representations separately. In §3.1, we define the NLD distance based on the Contextual Stochastic Block Model (CSBM) and verify its direct proportion to representation expressiveness. Due to the proved superiority of adaptive filters in heterophilic graphs[4, 11, 40], we approach the problem in the spectral domain. Specifically, we first explain the feasibility of acquiring NLD given the ego graph of a node in the spectral domain in §3.2. Then, we distill the NLD of nodes from filter performance through the bi-level optimization process, as spectral filter performance depends on the concentration of spectral label distribution [27, 9, 7]. In a nutshell, BioGNN distinguishes nodes with similar NLD but likely belong to different classes and feeds them into separate filters to prevent "loss rivalry". Our code is available at https://anonymous.4open.science/r/BioGNN-12B4.

**Our contributions.** (1) We reveal the "loss rivalry" phenomenon, where nodes belonging to different classes but with similar NLD tend to have opposite loss curves, which can negatively impact model convergence. (2) We provide theoretical explanations regarding the importance of NLD and the benefits of using polynomial-based spectral filtering methods to capture the NLD of nodes. (3) We propose a novel bi-level optimization framework to address the problem, and the effectiveness of the proposed method is verified through experiments.

## 2 PRELIMINARIES AND NOTATIONS

In GAD, anomalous and normal nodes can be modeled as an attributed graph $\mathcal{G} = (\mathcal{V}, \mathcal{E}, \mathbf{X})$, where $\mathcal{V}$ represents the set of anomalous and normal nodes, $\mathcal{E}$ denotes edges, and $\mathbf{X}$ is the attribute matrix. The objective of GAD is to identify anomalous nodes by learning

from the attributes and structure of the graph. In §3.1, we will discuss the impact of NLD on GAD and demonstrate the superiority of spectral filtering in addressing this issue. Therefore, we introduce basic knowledge of graph spectral filtering in this section.

**Graph-based Anomaly Detection.** A primary approach for GAD is to frame it as a semi-supervised node classification task [32]. The goal is to train a predictive GNN model $g$ that achieves minimal error in approaching the ground truth $\mathbf{Y}_{test}$ for unobserved nodes $\mathcal{V}_{test}$ given observed nodes $\mathcal{V}_{train}$, where $\mathcal{V}_{train} \cup \mathcal{V}_{test} = \mathcal{V}$ and $\mathcal{V}_{train} \cap \mathcal{V}_{test} = \emptyset$:

$$g(\mathcal{G}, \mathbf{Y}_{train}) \to \hat{\mathbf{Y}}_{test}. \tag{1}$$

Note that GAD is an imbalanced classification problem, which often results in similar NLD for normal nodes and anomalies: anomalies in the graph are rare, hence both normal nodes and anomalies are surrounded by numerous normal nodes.

**Graph Spectral Filtering.** Let $\mathbf{A}$ be the adjacency matrix, and $\mathcal{L}$ be the graph Laplacian, which can be expressed as $\mathbf{D} - \mathbf{A}$ or as $\mathbf{I} - \mathbf{D}^{-1/2}\mathbf{A}\mathbf{D}^{-1/2}$ (symmetric normalized), where $\mathbf{I}$ is the identity matrix, and $\mathbf{D}$ is the diagonal degree matrix. $\mathcal{L}$ is positive semi-definite and symmetric, so it has an eigen-decomposition $\mathcal{L} = \mathbf{U}\Lambda\mathbf{U}^T$, where $\Lambda = \{\lambda_1, \cdots, \lambda_N\}$ are eigenvalues, and $\mathbf{U} = [\mathbf{u}_1, \cdots, \mathbf{u}_N]$ are corresponding unit eigenvectors [40]. Assuming $\mathbf{X} = [\mathbf{x}_1, \cdots, \mathbf{x}_N]$ is a graph signal, we call the spectrum $\mathbf{U}^T\mathbf{X}$ the graph Fourier transform of the signal $\mathbf{X}$ [9, 44]. In graph signal processing (GSP), the frequency is associated with $\Lambda$. Therefore, the goal of spectral methods is to identify a response function $g(\cdot)$ on $\Lambda$ to learn the graph representation $\mathbf{Z}$ [8]:

$$\mathbf{Z} = g(\mathcal{L})\mathbf{X} = \mathbf{U}[g(\Lambda) \odot (\mathbf{U}^T\mathbf{X})] = \mathbf{U}g(\Lambda)\mathbf{U}^T\mathbf{X}. \tag{2}$$

## 3 THEORETICAL ANALYSIS

In this section, we introduce the Contextual Stochastic Block Model (CSBM), a widely used model for describing node feature formation. Based on CSBM, we define the NLD distance and verify its direct proportion to representation expressiveness. Furthermore, because adaptive filters have been shown to perform better in heterophilic graphs, we explore the feasibility of expressing NLD in the spectral domain to facilitate further study in later sections.

## 3.1 Impact of NLD on Node classification

GNNs are widely used to learn node representations in networks, as they can capture graph topological and structural information effectively. However, GNNs distinguish nodes by averaging the node features of their neighborhood [48]. Therefore, it is intuitive that the neighbor label distribution has a significant impact on GNN performance. To analyze NLD from a graph generation perspective, we introduce the Contextual Stochastic Block Model (CSBM) [14]. CSBM is a random graph generative model commonly used to measure the expressiveness of GNNs [33].

**CSBM.** The Contextual Stochastic Block Model (CSBM) makes the following assumptions for an attributed graph $\mathcal{G}$: (1) For a central node $u$ with label $c \in \{0, 1\}$, the labels of its neighbors are independently sampled from a fixed distribution $\mathcal{D}_c \sim Bern(p_c)$. $p_c$ denotes the sampling probability of class $c$, and the sampling process continues until the number of neighbors matches the degree of node $u$. In this work, we refer to the distribution $\mathcal{D}_c$ as the **neighborhood label distribution (NLD)**. (2) Anomalies and normal nodes have distinct node feature distributions, namely $\mathcal{F}_c$.

For simplicity, we define the NLD distance as follows:

**Definition 3.1** (Neighborhood Label Distribution Distance) Given a graph $\mathcal{G}$ with label vector $\mathbf{y}$, the neighborhood label distribution distance between nodes $u$ and $v$ is:

$$d(u, v) = dis(\mathcal{D}_{u_c}(u), \mathcal{D}_{v_c}(v)), \qquad (3)$$

where $dis(\cdot, \cdot)$ measures the difference between distribution vectors, such as cosine distance or Euclidean distance; $u_c$ and $v_c$ denote the class of nodes $u$ and $v$, respectively.

In this work, we focus on the binary GAD classification problem, hence $\mathcal{D}_c = \{\mathcal{D}_0 = [p_0, q_0], \mathcal{D}_1 = [p_1, q_1]\}$, where the symbol definitions are shown in Table 1. Furthermore, following previous works [6, 15, 33], we suppose that $\mathcal{F}_c$ are two Gaussian distributions of $n$ variables, i.e., $x_0 \sim N_n(\mu_0, \sigma^2 \mathbf{I})$, $x_1 \sim N_n(\mu_1, \sigma^2 \mathbf{I})$. This problem setting leads us to the following proposition, which indicates the expressive power of GNNs.

**Proposition 3.1** Given a graph $\mathcal{G} = (\mathcal{V}, \mathcal{E}, \{\mathcal{F}_c\}, \{\mathcal{D}_c\})$, the distance between the means of the class-wise hidden representations is proportional to their NLD distance.

**Remark.** The detailed proof can be found in Appendix A.1. This proposition shows that the expressive power of the representation depends on the neighborhood label distribution. Specifically, for nodes $u$ and $v$ in different classes, a vanilla 2-layer GCN has the following distance between their hidden representations:

$$||\mu_u - \mu_v||_2 = \frac{[d(u,v)]^2}{2} \cdot ||\mu_1 - \mu_0||_2, \qquad (4)$$

where $\mu_u$ and $\mu_v$ are the mean values of the learned representations of nodes $u$ and $v$. Similarly, for spectral methods, whose general polynomial approximation form can be written as $\sum_k \alpha_k \tilde{\mathcal{L}}^k \mathbf{X}$ [48], we can achieve a much larger NLD distance with a second-order polynomial:

$$||\mu_u - \mu_v||_2 = [1 + \frac{[d(u,v)]}{\sqrt{2}} + \frac{[d(u,v)]^2}{2}] \cdot ||\mu_1 - \mu_0||_2. \qquad (5)$$

**Table 1: NLD Symbol Definition.**

| Symbol | Definition (Probability of) |
|:---:|:---:|
| $p_1$ | normal neighbor for anomalies |
| $q_1$ | anomaly neighbor for anomalies |
| $p_0$ | normal neighbor for normal nodes |
| $q_0$ | anomaly neighbor for normal nodes |

The larger the distance $||\mu_u - \mu_v||_2$, the more expressive the representation and the better capability of the downstream linear detector. From (4) and (5), we observe two things: (1) the minimum value of $||\mu_u - \mu_v||_2$ is achieved when $d(u, v)$ is 0; (2) using second-order polynomial graph filtering can improve the ability to distinguish between nodes, especially when the NLD of nodes from different classes are similar. This finding aligns with previous research [49, 50] in this area.

## 3.2 NLD in the Spectral Domain

The NLD of anomalous and normal nodes in four benchmark datasets is statistically reported in Table 2. We observe that the NLD for nodes from different classes are similar, especially in YelpChi and Amazon datasets. Our analysis justifies the need to filter out anomalies sharing similar neighborhood labels with normal nodes, so that the distribution of the remaining anomalies can be distinguished from that of normal nodes. Proposition 3.1 suggests that spectral methods are more effective. Therefore, we aim to address the problem in the spectral domain. To begin with, we express NLD in the spectral domain by bridging the gap between it and frequency. Specifically, we fragment a graph into a set of ego-graphs [46] and define the spectral label distribution as follows:

**Definition 3.2** (Spectral Label Energy Distribution) Given an ego node $u$ and its one-hop neighbor set $\mathcal{N}_u$ with size $N$, the spectral label energy distribution at $\lambda_k$ is:

$$f_k(\mathbf{y}, \mathcal{L}) = \alpha_k^2 / \sum_{n=1}^N \alpha_i^2, \qquad (6)$$

where $f$ is a probability distribution with $\sum_{k=1}^N f_k = 1$, $\mathcal{L}$ is the Laplacian matrix of the ego-graph, and $\{\alpha\}$ denotes the ego-graph spectrum of the one-hot label vector $\mathbf{y}$. Since $\alpha_k = \mathbf{u}_k^T \mathbf{y}$, $f_k(\mathbf{y}, \mathcal{L})$ measures the weight of $\mathbf{u}_k$ in $\mathbf{y}$, a larger $f_k$ indicates that the spectral label distribution concentrates more on $\lambda_k$. With Definition 3.2, we now show the relationship between $f(\mathbf{y}, \mathcal{L})$ and NLD.

**Proposition 3.2** For a binary classification problem, the expectation of the spectral label energy distribution $\mathbb{E}[f(\mathbf{y}, \mathcal{L})]$ is positively associated with the NLD of the node. Specifically:

$$\mathbb{E}[f(\mathbf{y}, \mathcal{L})] = \begin{cases} \dfrac{|\mathcal{E}| \cdot (1 - p_0)}{N} & y = 0, \\[3mm] \dfrac{|\mathcal{E}| \cdot p_1}{N} & y = 1. \end{cases} \qquad (7)$$

**Remark.** The detailed proof can be found in Appendix A.2. Proposition 3.2 indicates that capturing the difference in spectral label distribution is equivalent to measuring the similarity between NLDs. Furthermore, the proposition elucidates that different nodes with

**Table 2: Summary of the dataset statistics and the neighbor label distributions.**

| Dataset | Statistics | | | Neighbor Label Distribution (NLD) | | | | |
|---------|-----------|--------|-----------|--------|--------|--------|--------|----------|
| | # Nodes | # Edges | # Features | $p_0$ | $q_0$ | $p_1$ | $q_1$ | Distance |
| YelpChi | 11,944 | 4,398,392 | 25 | 0.8683 | 0.1317 | 0.8144 | 0.1856 | 0.0762 |
| Amazon | 45,954 | 3,846,979 | 32 | 0.9766 | 0.0234 | 0.9254 | 0.0746 | 0.0724 |
| T-Finance | 39,357 | 21,222,543 | 10 | 0.9850 | 0.0150 | 0.5280 | 0.4720 | 0.6462 |
| T-Social | 5,781,065 | 73,105,508 | 10 | 0.7634 | 0.2366 | 0.9161 | 0.0839 | 0.2159 |

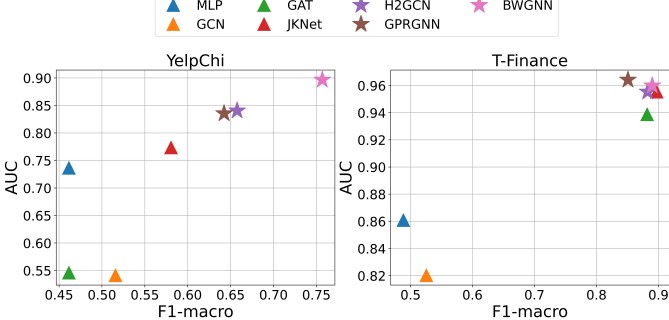

**Figure 3: The influence of NLD on model performance.**

similar NLD retain rather different frequency components. Based on this finding, separating nodes whose spectral label distributions are different could bring two benefits: (1) separate nodes in the same class but have different NLDs; (2) separate nodes in different classes but have similar NLDs. Both of these benefits alleviate the "loss rivalry" phenomenon and help with the convergence of GNNs.

## 3.3 Validation on Real-World Graphs

To verify the correctness of our theoretical findings, we report the F1-macro and AUC performance of some general methods (triangle marker) [26, 41, 49] and some polynomial spectral methods [11, 40, 50] (star marker) in Figure 3. We make two observations: (1) As shown in Table 2, the NLD distance between the two classes is 0.0762 and 0.6462 for YelpChi and T-Finance, respectively. From Figure 3, we observe that most methods achieve better results on T-Finance than on YelpChi, demonstrating the importance of NLD. Moreover, the performance gap between models on YelpChi and Amazon is much larger than that on T-Finance. This suggests that we can achieve decent performance with less powerful models on datasets with larger NLD distances. Our finding supports the notion that NLD can influence the expressive power of the GNN model, and separating nodes with specific NLDs can improve the performance of the GNN model. (2) Spectral methods outperform spatial methods by a large margin. These tailored heterophilic filters further support our argument for the superiority of addressing the problem in the spectral domain.

## 4 METHODOLOGY

Guided by the analysis in §3.1, we advocate for the necessity of treating nodes with distinct spectral label distributions separately. In this section, we introduce our **bi**-level **o**ptimization **g**raph **n**eural

network BioGNN. To begin with, we introduce the learning objectives in §4.1 and present the parameterization process in §4.2. In §4.4, we validate the effectiveness of the framework on golden-separated graphs.

## 4.1 The Learning Objectives

To start with, we introduce Lemma 4.1 which is widely agreed upon in the literature [7, 9, 27]:

**Lemma 4.1** The prediction performance of a spectral filter is better when the spectral label energy distribution concentrates more on the pass band of the filter.

Building on Lemma 4.1, we could identify nodes according to the performance of different spectral filters through bi-level optimization. As shown in Figure 4, our learning objective is twofold: (1) Optimize the encoders $\{\Phi(\cdot), \varphi_1(\cdot), \varphi_2(\cdot)\}$ to maximize the probability of correctly classifying nodes separated by $\theta(\cdot)$; (2) Optimize the encoder $\theta(\cdot)$ which predicts the NLD of nodes and separate nodes to two sets. We set all the encoders as MLP with learnable parameters. Concretely, the learning objective of BioGNN is defined as follows:

$$\min_{\varphi_1, \Phi, M_1} \mathcal{R}(\Phi(g_1(\mathcal{L})\varphi_1(M_1 \circ \mathbf{X})), \mathbf{Y})$$
$$+ \min_{\varphi_2, \Phi, M_2} \mathcal{R}(\Phi(g_2(\mathcal{L})\varphi_2(M_2 \circ \mathbf{X})), \mathbf{Y}), \qquad (8)$$
$$s.t. \quad M_1 + M_2 = \mathbf{1},$$

where $M_1$ and $M_2$ are hard masks given by learnable encoder $\theta(\cdot)$, **1** is an all-one vector, $g_1(L)$ and $g_2(L)$ are spectral filters, and $\circ$ denotes the element-wise multiplication.

## 4.2 Instantiation of BioGNN

Given the two-fold objective, we propose to parameterize the encoder $\theta(\cdot)$ and $\{\Phi(\cdot), \varphi_1(\cdot), \varphi_2(\cdot)\}$.

*Parameterizing $\theta(\cdot)$.* The encoder $\theta(\cdot)$ serves as a separator that predicts the NLD of nodes and feeds nodes into different branches of filters. Consequently, to obtain informative input for $\theta$, we employ a label-wise message passing layer [12] which aggregates the labeled neighbors of the nodes label-wise. Concretely, for node $u$, the aggregated feature $h_{u,c}$ for class $c$:

$$h_{u,c} = \frac{1}{|\mathcal{N}_{l,c}(u)|} \sum_{v \in \mathcal{N}_{l,c}(u)} x_v, \qquad (9)$$

where $\mathcal{N}_{l,c}(u)$ is the set of neighbors labeled with $c$. When there are no labeled neighbors belonging to class $c$, we assign a zero

**Figure 4: BioGNN Framework. Mask generator $\theta(\cdot)$ identifies subsets of nodes according to equation (10). Two projection heads $\varphi_1(\cdot)$ and $\varphi_2(\cdot)$ and two spectral filters $g_1(L)$ and $g_2(L)$ assign labels to according subset of nodes. Mask generator and filters are optimized iteratively according to equation (11) and equation (12).**

embedding to $h_{u,c}$. Then we set

$$M_1(u) = \arg\max(\text{MLP}_\theta([x_u; h_{u,0}; h_{u,1}])). \tag{10}$$

To ensure smoothed and well-defined gradients $\frac{\partial y}{\partial \theta}$, we apply a straight-through (ST) gradient estimator [2] to make the model differentiable. Note that BioGNN is trained in an iterative fashion, the encoders $\{\Phi(\cdot), \varphi_1(\cdot), \varphi_2(\cdot)\}$ are fixed as $\{\Phi^*(\cdot), \varphi_1^*(\cdot), \varphi_2^*(\cdot)\}$, the objective function in this phase is:

$$\min_{M_1} \mathcal{R}(\Phi^*(g_1(\mathcal{L})\varphi_1^*(M_1 \circ \mathbf{X})), \mathbf{Y})$$
$$+ \min_{M_2} \mathcal{R}(\Phi^*(g_2(\mathcal{L})\varphi_2^*(M_2 \circ \mathbf{X})), \mathbf{Y}) \tag{11}$$
$$s.t. \quad M_1 + M_2 = \mathbf{1}.$$

*Parameterizing* $\{\Phi(\cdot), \varphi_1(\cdot), \varphi_2(\cdot)\}$. These three encoders serve as a predictor that assigns labels to input nodes. As we aim to distinguish between different spectral label distributions, which are closely related to the performance of filters with corresponding band-pass, we adopt low-pass and high-pass filters as $g_1(L)$ and $g_2(L)$, respectively. Here, we choose to use two branches and leave the multi-branch framework for future work. Therefore, the functions of $M_1$ and $M_2$ become the masking of nodes with high-frequency and low-frequency ego-graphs, respectively. In this iterative training phase, we freeze the masks as $M_1^*$ and $1 - M_1^*$, and set the objective function as:

$$\min_{\Phi, \varphi_1} \mathcal{R}(\Phi(g_1(\mathcal{L})\varphi_1(M_1^* \circ \mathbf{X})), \mathbf{Y})$$
$$+ \min_{\Phi, \varphi_2} \mathcal{R}(\Phi(g_2(\mathcal{L})\varphi_2((1 - M_1^*) \circ \mathbf{X})), \mathbf{Y}). \tag{12}$$

A similar training process has also been used in graph contrastive learning [39]. For the choice of $g_1(\mathcal{L})$ and $g_2(\mathcal{L})$, we adopt Bernstein polynomial-based filters [23, 40] for their convenience to decompose low-pass and high-pass filters:

$$g(\mathcal{L}) = \frac{1}{2} U\beta_{\alpha,\beta}(\Lambda)U^T = \frac{(\mathcal{L}/2)^\alpha(I - \mathcal{L}/2)^\beta}{2\int_0^1 t^{\alpha-1}(1-t)^{\beta-1}dt}, \tag{13}$$

where $\beta_{\alpha,\beta}$ is the standard beta distribution parameterized by $\alpha$ and $\beta$. When $\alpha \to 0$, we acquire $g(\mathcal{L})$ as a low-pass filter; similarly, $g(\mathcal{L})$ acts as a high-pass filter when $\beta \to 0$. For the choices of $\alpha$ and $\beta$ on the specific benchmark and more training details, please refer to Appendix B.1 and B.2.

## 4.3 Initialization of BioGNN

To embrace a more stable process of the bi-level optimization, we initialize the encoders before iterative training.

**Initialization of $\theta(\cdot)$.** $\theta(\cdot)$ is initialized in a supervised fashion, where the supervision signal is obtained by counting the labeled inter-class neighbors:

$$Y_{sep}(u) = round(\frac{1}{|\mathcal{N}_L(u)|} \sum_{v \in \mathcal{N}_L(u)} |\{y_u \neq y_v\}|), \tag{14}$$

then the cross-entropy is minimized:

$$\min_\theta -[\mathbf{Y}_{sep} \circ log(\theta(\mathbf{X})) + (1 - \mathbf{Y}_{sep}) \circ log(1 - \theta(\mathbf{X}))]. \tag{15}$$

Note that in our experiments, although the supervision signal are calculated with ego-graphs, the input data is a **complete graph** rather than ego-graphs extracted from a larger graph. Each node in the complete graph connects directly to all other nodes, ensuring that all interactions are considered during the learning process. As nodes with high-frequency ego-graph are rare, to shield the separator from predicting all nodes as low-frequency ego nodes, we regularize the ratio of two sets of nodes by enforcing the following constraint: we treat $\mathbf{Y}_{sep}$ as the optimal known mask, and one term $\mathbf{Y}_{sep} - \theta(\mathbf{X})$ is added to the objective. The final objective is:

$$\min_\theta -[\mathbf{Y}_{sep} \circ log(\theta(\mathbf{X})) + (1 - \mathbf{Y}_{sep}) \circ log(1 - \theta(\mathbf{X}))] + \gamma(\mathbf{Y}_{sep} - \theta(\mathbf{X})). \tag{16}$$

**Initialization of $\{\Phi(\cdot), \varphi_1(\cdot), \varphi_2(\cdot)\}$.** In this phase, we treat $Y_{sep}$ as the optimal known mask:

$$\min_{\Phi, \varphi_1} \mathcal{R}(\Phi(g_1(\mathcal{L})\varphi_1(Y_{sep} \circ \mathbf{X})), \mathbf{Y})$$
$$+ \min_{\Phi, \varphi_2} \mathcal{R}(\Phi(g_2(\mathcal{L})\varphi_2((1 - Y_{sep}) \circ \mathbf{X})), \mathbf{Y}). \tag{17}$$

## 4.4 Validation on Golden-separated Graphs

From an omniscient perspective, we can validate the effectiveness of BioGNN. Assuming that we know all the labels of the nodes, we have access to the accurate NLD of all the nodes. In this case, we can separate the nodes ideally, *i.e.,* , we train them separately in sequence.

From Figure 5a, we observe that the loss decreases smoothly, demonstrating our argument that mixed nodes are the main cause of the "loss rivalry" phenomenon. Based on this finding, BioGNN

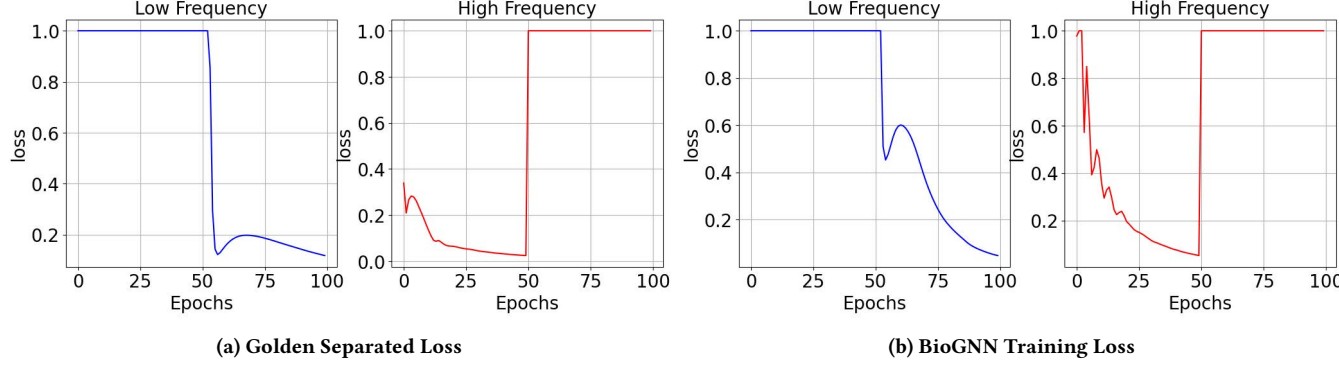

(a) Golden Separated Loss

(b) BioGNN Training Loss

**Figure 5: Golden separated loss and real loss curves on YelpChi.**

can alleviate the problem and boost the performance of GAD. We discovered that the training order is significant in achieving better performance. Training nodes with high-frequency ego-graphs before those with low-frequency ones leads to better results. One possible reason for this is the shared linear classifier $\Phi$ between the two branches. Embeddings learned from the high-pass filter are noisier, and a classifier that performs well on noisy embeddings would most likely perform well on the whole dataset [24]. We consider this to be an intriguing discovery, yet leaving a comprehensive theoretical examination for future research.

## 5 EXPERIMENT

In this section, we conduct experiments on four benchmarks and report the results of our models as well as some state-of-the-art baselines to demonstrate the effectiveness of BioGNN.

### 5.1 Experimental Setup

**Datasets.** Following previous works [18, 40], we conduct experiments on four datasets introduced in Table 2. For more details about the datasets, please refer to Appendix B.3.

**Baselines.** Our baselines can be roughly categorized into three groups. The first group includes general methods, such as **MLP**, **GCN** [26], **GAT** [41], **ChebyNet** [13], **GWNN** [47], and **JKNet** [49]. As our focus is GAD, the second group considers tailored GAD methods including **CAREGNN** [18], **PCGNN** [30], **GDN** [20], and **BWGNN** [40]. The third group includes methods that consider neighbor labels, such as **H2GCN** [50], **GPRGNN** [11], and **MixHop** [1]:

- **GCN** [26]: GCN is a traditional graph convolutional network in spectral space.
- **GAT** [41]: GAT leverages masked self-attentional layers to weight the neighbors.
- **ChebyNet** [13]: ChebyNet generalizes CNN to graph data in the context of spectral graph theory.
- **GWNN** [47][1]: GWNN leverages graph wavelet transform to address the shortcomings of spectral graph CNN methods that depend on graph Fourier transform.

- **JKNet** [49]: The jumping-knowledge network which concatenates or max-pooling the hidden representations.
- **Care-GNN** [18] [2]: Care-GNN is a camouflage-resistant graph neural network that adaptively samples neighbors according to the feature similarity, and the optimal sampling ratio is found through an RL module.
- **PC-GNN** [30] [3]: PC-GNN consists of two modules "pick" and "choose", and maintains a balanced label frequency around fraudsters by downsampling and upsampling.
- **H2GCN** [50] [4]: H2GCN is a tailored heterophily GNN which identifies three useful designs.
- **BWGNN** [40] [5]: BWGNN is a spectral filter addressing the "right-shift" phenomenon in anomaly detection.
- **GDN** [20] [6]: GDN deals with heterophily by leveraging constraints on original node features.
- **MixHop** [1] [7]: Mixhop repeatedly mixes feature representations of neighbors at various distances to learn relationships.
- **GPRGNN** [11] [8]: GPR-GNN learns a polynomial filter by directly performing gradient descent on the polynomial coefficients.

### 5.2 Performance Comparison

The main results are reported in Table 3. Note that we search for the best threshold to achieve the best F1-macro in validation for all methods. In general, BioGNN achieves the best F1-macro score in all datasets, empirically verifying that it has a larger distance between predictions and the decision boundary, benefiting from measuring the NLD distance. For AUC, BioGNN did not achieve the best score in T-Social. We suppose the reason is that T-social has a complex frequency composition since the best performance is achieved when the frequency order is high according to BWGNN [40]. We believe this issue could be alleviated if multi-branch filters are adopted, which we leave for future work. Furthermore, some methods could achieve high AUC while maintaining a low F1-Macro, indicating that the instances can be distinguished but hold tightly in the space.

---

[1]https://github.com/benedekrozemberczki/GraphWaveletNeuralNetwork

[2]https://github.com/YingtongDou/CARE-GNN
[3]https://github.com/PonderLY/PC-GNN
[4]https://github.com/GemsLab/H2GCN
[5]https://github.com/squareRoot3/Rethinking-Anomaly-Detection
[6]https://github.com/blacksingular/wsdm_GDN
[7]https://github.com/samihaija/mixhop
[8]https://github.com/jianhao2016/GPRGNN

**Table 3: Performance Results. The best results are in boldface, and the 2nd-best are underlined.**

| Dataset | YelpChi | | Amazon | | T-Finance | | T-Social | |
|---------|---------|-----|--------|-----|-----------|-----|----------|-----|
| Metric | F1-Macro | AUC | F1-Macro | AUC | F1-Macro | AUC | F1-Macro | AUC |
| MLP | 0.4614 | 0.7366 | 0.9010 | 0.9082 | 0.4883 | 0.8609 | 0.4406 | 0.4923 |
| GCN | 0.5157 | 0.5413 | 0.5098 | 0.5083 | 0.5254 | 0.8203 | 0.6550 | 0.7012 |
| GAT | 0.4614 | 0.5459 | 0.5675 | 0.7731 | 0.8816 | 0.9388 | 0.4921 | 0.4923 |
| ChebyNet | 0.4608 | 0.6216 | 0.8070 | 0.9187 | 0.8017 | 0.8001 | OOM | |
| GWNN | 0.4608 | 0.6246 | 0.4822 | 0.9319 | 0.4883 | 0.9670 | OOM | |
| JKNet | 0.5805 | 0.7736 | 0.8270 | 0.8970 | 0.8971 | 0.9554 | 0.4923 | 0.7226 |
| CAREGNN | 0.5015 | 0.7300 | 0.6313 | 0.8832 | 0.7261 | 0.9105 | 0.4868 | 0.7939 |
| PCGNN | 0.6925 | 0.8118 | 0.8367 | 0.9555 | 0.4462 | 0.9200 | 0.4536 | 0.8917 |
| GDN | 0.7545 | 0.8904 | 0.9068 | 0.9709 | 0.8474 | 0.9462 | 0.7401 | 0.9287 |
| BWGNN | 0.7568 | **0.8967** | 0.9204 | 0.9706 | 0.8899 | 0.9599 | 0.7494 | 0.9275 |
| H2GCN | 0.6575 | 0.8406 | 0.9213 | 0.9693 | 0.8824 | 0.9553 | OOM | OOM |
| MixHop | 0.6534 | 0.8796 | 0.8093 | 0.9723 | 0.4880 | 0.9569 | 0.6471 | 0.9597 |
| GPRGNN | 0.6423 | 0.8355 | 0.8059 | 0.9358 | 0.8507 | 0.9642 | 0.5976 | **0.9622** |
| BioGNN | **0.7606** | 0.8947 | **0.9462** | **0.9766** | **0.9059** | **0.9670** | **0.8140** | 0.9325 |

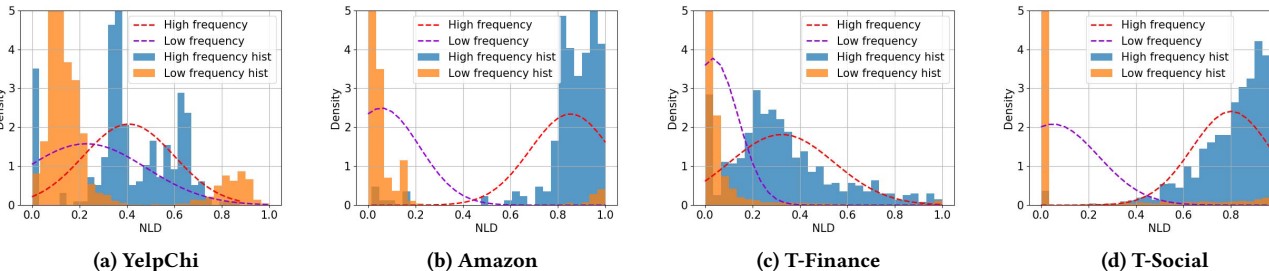

(a) YelpChi          (b) Amazon          (c) T-Finance          (d) T-Social

**Figure 6: The NLD distance between two separated sets of nodes.**

In such cases, it is hard to identify a classification threshold, which we consider unstable.

H2GCN, MixHop, and GPRGNN are three state-of-the-art spectral heterophilous GNNs that shed light on the relationship between the ego node and neighbor labels. We observe that they consistently outperform other groups of methods, including some tailored GAD methods. We ascribe this large performance gap to two reasons: (1) the harmfulness of heterophily where vast normal neighborhoods attenuate the suspiciousness of the anomalies; (2) the superiority of spectral filters to distinguish nodes with different NLD. However, they optimize the node representations as a whole, while BioGNN outperforms these methods, especially in F1-Macro, where the improvement ranges from 2.7% to 25.8%. This supports our analysis that different class nodes with similar NLD should be treated separately to alleviate "loss rivalry". Furthermore, among the tailored GNN methods (CAREGNN, PCGNN, GDN, BWGNN, and BioGNN), BWGNN and BioGNN are polynomial-based filters that perform better than others, further suggesting that spectral filtering is more promising in GAD.

In several datasets, MLP outperforms some GNN-based methods, indicating that blindly mixing neighbors can sometimes degrade the prediction performance. Therefore, structural information should be used with care, especially when the neighborhood label distributions for nodes are complex.

## 5.3 Analysis of BioGNN

In this section, we take a closer look in BioGNN. We first verify the smoothness of the BioGNN loss curve to demonstrate its effectiveness in alleviating "loss rivalry". Then we plot the distribution of the separated nodes to elucidate that our model can successfully discriminate nodes with different NLD and set them apart. Making it more clear, we visualize some high-frequency ego-graphs.

**Loss Rivalry Addressing.** To answer the question of whether BioGNN can alleviate the "loss rivalry", we plot the training loss of BioGNN in Figure 5b. Similar to Section 4.4, two separate sets of nodes are trained in a specific order: high-frequency nodes are trained first, followed by low-frequency nodes. Comparing Figure 2, 5a, and 5b, we find that the smoothness of BioGNN's training curve lies between golden-separate and mixed training, indicating that the new framework is effective in alleviating "loss rivalry" and improves the overall performance of GAD.

**Distribution of the separated nodes.** The core of BioGNN is node separation. To further validate its effectiveness, we report the

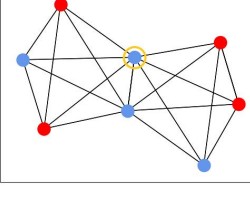 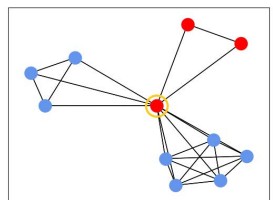 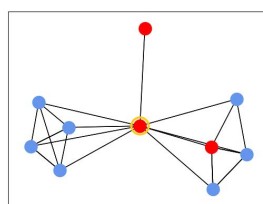 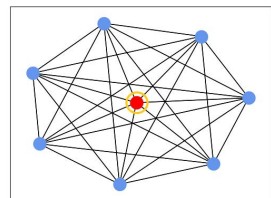

**Figure 7: The ego-graph of some yellow-circled ego nodes classified as high-frequency by BioGNN in YelpChi. The anomalies are represented in red, while normals are represented in blue.**

empirical histogram of the NLD in four benchmarks in Figure 6. The x-axis represents the edge homophily, which explicitly represents the NLD around the ego node. The y-axis denotes the density, and the distribution curves are shown in dashed lines. From Figure 6, we observe that the two histograms seldom overlap, and the mean of two curves maintains a separable distance, demonstrating that BioGNN successfully sets the nodes apart.

**Visualization.** To show the results in an intuitive way, we report the ego-graph of some nodes in Figure 7. These nodes are assigned to the high-pass filter by $\theta(\cdot)$. As observed from the figure where color denotes the class of the nodes, the ego node (red-circled) has more inter-class neighbors compared to the nodes assigned to the low-pass filter. This finding provides support for Equation 7 and verifies the effectiveness of our novel framework. More visualizations are in Appendix C.

**Time complexity analysis.** The time complexity of BioGNN is $O(C|\mathcal{E}|)$, where $C$ represents a constant and $|\mathcal{E}|$ denotes the number of edges in the graph. This is due to the fact that the BernNet-based filter is a polynomial function that can be computed recursively, as explained in [40].

## 6 RELATED WORK

In this section, we introduce some static GAD networks and polynomial-based spectral GNNs.

### 6.1 Static Graph Anomaly Detection

On static attributed graphs, GNN-based semi-supervised learning methods are widely adopted. For example, GraphUCB [16] adopts contextual multi-armed bandit technology, and transforms graph anomaly detection into a decision-making problem; DCI [45] decouples representation learning and classification with the self-supervised learning task. Recent methods realize the necessity of leveraging multi-relation graphs into GAD. FdGars [43] and Graph-Consis [31] construct a single homo-graph with multiple relations. Likewise, Semi-GNN [42], CARE-GNN [18], and PC-GNN [30] construct multiple homo-graphs based on node relations. In addition, some works discover that heterophily should be addressed properly in GAD. Semi-GNN and IHGAT [28] employ hierarchical attention mechanisms for interpretable prediction, while based on camouflage behaviors and imbalanced problems, CARE-GNN, PC-GNN, and AO-GNN [25] prune edges adaptively according to neighbor

distribution. GDN [21] and H2-FDetector [38] adopt different strategies for anomalies and normal nodes.

### 6.2 Graph Spectral Filtering

Spectral GNNs simulate filters with different passbands in the spectral domain, enabling GNNs to work on both homophilic and heterophilic graphs [44]. GPRGNN [11] adaptively learns the Generalized PageRank weights, regardless of whether the node labels are homophilic or heterophilic. FSGNN [34] designs a feature selection graph neural network. FAGCN [3] adaptively fuses different signals in the process of message passing by employing a self-gating mechanism. BernNet [23] expresses the filtering operation with Bernstein polynomials. BWGNN [40] observes the "right-shift" phenomenon and designs a band-pass filter to aggregate different frequency signals simultaneously. AdaGNN [17] captures the varying importance of different frequency components to alleviate over-smoothing problem. AMNet [5] aims to capture both low-frequency and high-frequency signals, and adaptively combine signals of different frequencies. GHRN [19] design an edge indicator to distinguish homophilous and heterophilous edges.

## 7 LIMITATION AND CONCLUSION

**Limitation.** Although we propose a novel network that treats nodes separately, it has some limitations. Our work only separates the nodes into two sets, and we hope to extend it to more fine-grained multi-branch neural networks in the future. Furthermore, our theoretical result largely relies on CSBM's assumptions; hence our model may fail in some cases where the graph generation process doesn't follow these assumptions.

**Conclusion.** This work starts with "loss rivalry", expressing the phenomenon that some nodes tend to have opposite loss curves from others. We argue that it is caused by the mixed training of different class nodes with similar NLD. Furthermore, we discover that spectral filters are superior in addressing the problem. To this end, we propose BioGNN, which essentially discriminates nodes that share similar NLD but are likely to be in different classes and feeds them into different filters to prevent "loss rivalry".

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

## A PROOFS

In this section, the proofs of propositions are listed.

### A.1 Proof of Proposition 1

*Proof.* In the spectral domain, the hidden representation of the spectral filter can be expressed as:

$$H = \sum_k \alpha_k \tilde{\mathcal{L}}^k \mathbf{X} = \sum_k \alpha_k (\mathbf{I} - \mathbf{D}^{-1/2}\mathbf{A}\mathbf{D}^{-1/2})^k \mathbf{X} \qquad (18)$$

Taking the second-order spectral filter as an example,

$$H^2 = \alpha_0 \mathbf{X} + \alpha_1 (\mathbf{I} - \mathbf{D}^{-1/2}\mathbf{A}\mathbf{D}^{-1/2})\mathbf{X} + \alpha_2 (\mathbf{I} - \mathbf{D}^{-1/2}\mathbf{A}\mathbf{D}^{-1/2})^2 \mathbf{X} \quad (19)$$

The representation of node $i$ is given as:

$$h_i = \alpha_0 x_i + \alpha_1 (x_i - \frac{1}{deg(x_i)} \sum_{j \in \mathcal{N}_i} x_j) +$$

$$\alpha_2 (x_i - 2\frac{1}{deg(x_i)} \sum_{j \in \mathcal{N}_i} x_j + \frac{1}{deg(x_i)} \sum_{j \in \mathcal{N}_i} \frac{1}{deg(x_j)} \sum_{k \in \mathcal{N}_j} x_k)$$

$$= (\alpha_0 + \alpha_1 + \alpha_2) x_i - \frac{\alpha_1 + 2\alpha_2}{deg(x_i)} \sum_{j \in \mathcal{N}_i} x_j$$

$$+ \frac{\alpha_2}{deg(x_i)} \sum_{j \in \mathcal{N}_i} \frac{1}{deg(x_j)} \sum_{k \in \mathcal{N}_j} x_k \qquad (20)$$

Here we only focus on the aggregation process, hence the non-linear activation is ignored. Furthermore, to simplify the calculation and the analysis, we set $\alpha_0$ as 1, $\alpha_1$ as -1, and $\alpha_2$ as 1. In this case the coefficients or the numerators of the coefficients equal to 1. Suppose $u$ and $v$ are nodes with different labels (*i.e.*, anomalies and normal nodes), along with their NLD as $\mathcal{D}_u = [p_1, q_1]$ and $\mathcal{D}_v = [p_0, q_0]$, where $p_1 + q_1 = p_0 + q_0 = 1$. From previous analysis, we assume $x_u \sim N(\mu_1, \mathbf{I})$ and $x_v \sim N(\mu_0, \mathbf{I})$, hence we know $h_u$ and $h_v$ should obey Gaussian distribution, whose mean can be acquired as:

$$\mu_u = \mu_1 - (p_1\mu_0 + q_1\mu_1) + p_1(p_0\mu_0 + q_0\mu_1) + q_1(p_1\mu_0 + q_1\mu_1)$$
$$= \mu_1 + p_1(p_0\mu_0 + q_0\mu_1 - p_1\mu_0 - q_1\mu_1)$$
$$= \mu_1 + p_1[(p_0 - p_1)\mu_0 + (q_0 - q_1)\mu_1]$$
$$\mu_v = \mu_0 - q_0[(p_0 - p_1)\mu_0 + (q_0 - q_1)\mu_1] \qquad (21)$$

Hence the distance between the mean of these two distributions is:

$$||\mu_u - \mu_v||_2 = ||\mu_1 - \mu_0||_2 + (p_1 + q_0)||(p_0 - p_1)\mu_0 + (q_0 - q_1)\mu_1||_2$$
$$= ||\mu_1 - \mu_0||_2 + (1 + q_0 - q_1) \cdot |q_0 - q_1| \cdot ||\mu_1 - \mu_0||_2$$
$$= [1 + |q_0 - q_1| + |(p_0 - p_1)(q_0 - q_1)|] \cdot ||\mu_1 - \mu_0||_2 \qquad (22)$$

Similarly, since $|q_0 - q_1| = |p_0 - p_1|$, we have:

$$||\mu_u - \mu_v||_2 = [1 + |p_0 - p_1| + |(p_0 - p_1)(q_0 - q_1)|] \cdot ||\mu_1 - \mu_0||_2 \quad (23)$$

In our paper, we adopt Euclidean distance between vectors as NLD:

$$d(u, v) = \sqrt{(p_0 - p_1)^2 + (q_0 - q_1)^2} \qquad (24)$$

Joining equations (23) and (24), we can rewrite the distance between distribution mean values as:

$$||\mu_u - \mu_v||_2 = [1 + \frac{[d(u,v)]}{\sqrt{2}} + \frac{[d(u,v)]^2}{2}] \cdot ||\mu_1 - \mu_0||_2. \qquad (25)$$

Likewise, the mean values of hidden represenation given by a 2-layer vanilla GCN are:

$$\mu_u = p_1(p_0\mu_0 + p_1\mu_1) + q_1(p_1\mu_0 + p_0\mu_1)$$
$$= \mu_0 + p_1^2(\mu_1 - \mu_0) + q_1 p_0(\mu_1 - \mu_0)$$
$$\mu_v = p_0(p_0\mu_0 + p_1\mu_1) + q_0(p_1\mu_0 + p_0\mu_1)$$
$$= \mu_0 + p_1 p_0(\mu_1 - \mu_0) + q_0 p_0(\mu_1 - \mu_0) \qquad (26)$$

Hence we have the distance between them:

$$||\mu_u - \mu_v||_2 = p_1 \cdot |p_1 - p_0| \cdot ||\mu_1 - \mu_0||_2 - p_0 \cdot |q_1 - q_0| \cdot ||\mu_1 - \mu_0||_2 \qquad (27)$$

Since $|p_1 - p_0| = |1 - q_1 - (1 - q_0)| = |q_0 - q_1|$

$$||\mu_u - \mu_v||_2 = |(p_0 - p_1)(q_0 - q_1)| \cdot ||\mu_1 - \mu_0||_2 \qquad (28)$$

Joining Equations (24) and (28), we can rewrite the distance as:

$$||\mu_u - \mu_v||_2 = \frac{[d(u,v)]^2}{2} \cdot ||\mu_1 - \mu_0||_2, \qquad (29)$$

Finish the proof.

### A.2 Proof of Proposition 2

The Rayleigh quotient is widely adopted as the smoothness index which plays the role of frequency in classical spectral analysis. Here we adopt this metric to bridge two variables. Specifically, the Rayleigh quotient of the one-hot label vector $\mathbf{y}$ is:

$$E[\mathbf{y}] = \mathbf{y}^T \mathcal{L} \mathbf{y} = \mathbf{y}^T \mathbf{D} \mathbf{y} - \mathbf{y}^T \mathbf{A} \mathbf{y} = \sum_{i=1}^N d_i \mathbf{y}_i^2 - \sum_{i,j=1}^N \mathbf{y}_i \mathbf{y}_j \mathbf{A}_{ij}$$
$$= \frac{1}{2}(\sum_{i=1}^N d_i \mathbf{y}_i^2 - 2\sum_{i,j=1}^N \mathbf{y}_i \mathbf{y}_j A_{ij} + \sum_{j=1}^N d_j \mathbf{y}_j^2)$$
$$= \frac{1}{2} \sum_{(i,j) \in \mathcal{E}} (\mathbf{y}_i - \mathbf{y}_j)^2$$
$$= \sum_{(i,j) \in \mathcal{E}} \mathbb{I}\{\mathbf{y}_i \neq \mathbf{y}_j\}$$
$$= |\mathcal{E}| \cdot (1 - h(\mathcal{G})) \qquad (30)$$

On the other hand, the Rayleigh quotient can also be acquired as:

$$E[\mathbf{y}] = \mathbf{y}^T \mathbf{U} \Lambda \mathbf{U}^T \mathbf{y} = \alpha^T \Lambda \alpha$$
$$= \sum_{i=1}^N \lambda_i \alpha_i^2$$
$$= \sum_{i=1}^N \alpha_i^2 \mathbb{E}[f(\mathbf{y}, \mathcal{L})]$$
$$= N \mathbb{E}[f(\mathbf{y}, \mathcal{L})] \qquad (31)$$

Joining Equations (30) and (31), we have:

$$\mathbb{E}[f(\mathbf{y}, \mathcal{L})] = \frac{|\mathcal{E}| \cdot (1 - h(\mathcal{G}))}{N} \qquad (32)$$

Table 4: Model Hyperparameters and their search ranges

| Dataset | YelpChi | Amazon | T-Finance | T-Social |
|---|---|---|---|---|
| $\alpha$ | {**0**,1,2} | {**0**,1,2} | {**0**,1,2} | {**0**,1,2} |
| $\beta$ | {0,**1**,2} | {0,**1**,2} | {0,**1**,2} | {0,1,**2**} |
| Learning Rate ($lr$) for $\theta$ | {1e-3, 5e-3, **1e-2**} | | | |
| Learning Rate ($lr$) for $\Phi$ | {1e-3, 5e-3, **1e-2**} | | | |
| Learning Rate ($lr$) for $\varphi_1$ and $\varphi_2$ | {**1e-3**, 5e-3, 1e-2} | | | |
| weight decay for $\Phi$ | 1e-3 | | | |

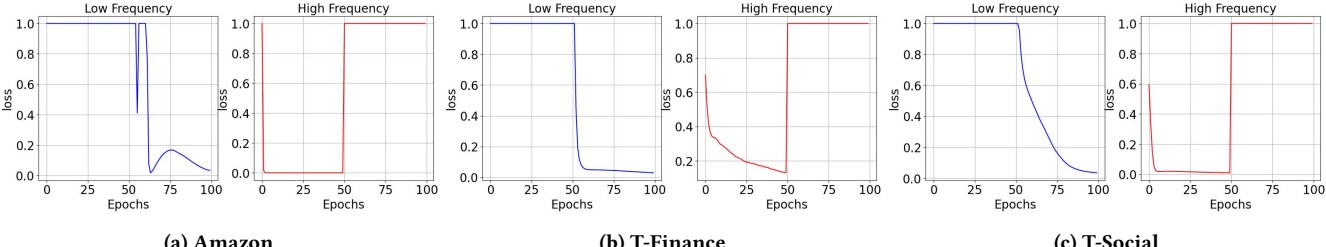

(a) Amazon          (b) T-Finance          (c) T-Social

Figure 8: More training curves of BioGNN.

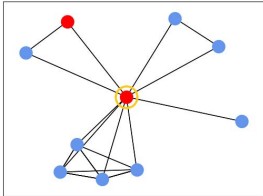 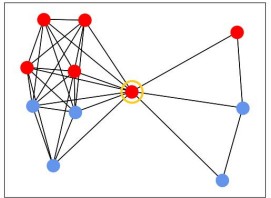 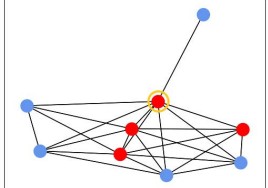 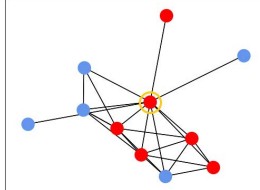

Figure 9: The ego-graph of some yellow-circled ego nodes classified as high-frequency by BioGNN in Amazon.

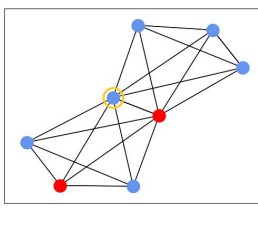 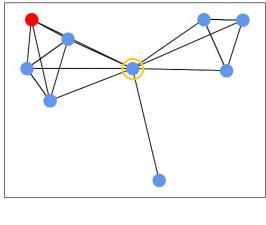 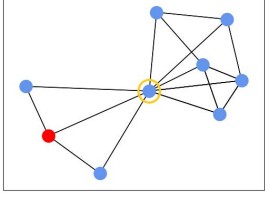 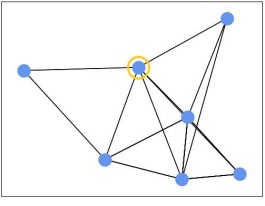

(a) yelp-1          (b) yelp-2          (c) amazon-1          (d) amazon-2

Figure 10: The ego-graph of some yellow-circled ego nodes classified as low-frequency by BioGNN.

Note that $\mathcal{G}$ is the ego-graph of the node, hence $1 - h(\mathcal{G})$ is the ratio of inter-class edges, which is $q_0$ for negative nodes and $p_1$ for positive nodes. Finish the proof.

## B REPRODUCIBILITY

In this section, some details for reproducibility are listed.

## Table 5: Performance with limited label information or (and) small percentage of abnormal nodes

| | BWGNN(F1-Macro %) | BWGNN(AUC %) | BioGNN(F1-Macro %) | BioGNN(AUC %) |
|---|---|---|---|---|
| Yelp (anomaly=5%, training=40%) | 76.44 | 89.67 | 74.71 | 88.17 |
| Yelp(anomaly=14.53%, training=1%) | 67.02 | 79.65 | 67.12 | 80.20 |
| Yelp (anomaly=5%, training=1%) | 66.27 | 78.49 | 64.93 | 74.57 |
| Amazon (anomaly=5%, training=40%) | 91.20 | 96.55 | 90.98 | 96.60 |
| Amazon (anomaly=6.87%, training=1%) | 90.69 | 91.24 | 84.36 | 92.46 |
| Amazon (anomaly=5%, training=5%) | 89.69 | 94.20 | 86.90 | 93.60 |
| TFinance (anomaly=4.58%, training=1%) | 84.89 | 91.15 | 83.14 | 92.53 |
| TSocial (anomaly=3.01%, training=1%) | 75.93 | 88.06 | 83.07 | 93.75 |

## B.1 Model Hyperparameters

According to [40], a Bernstein Polynominal-Based filter is parameterized by $\alpha$ and $\beta$. The choice of $\alpha$ and $\beta$ on datasets are presented in Table 4. In addition, some basic learning hyperparameters are reported.

## B.2 Datasets

The YelpChi dataset [36] focus on detecting anomalous recommendations from Yelp.com. The Amazon dataset [35] includes product reviews under the Musical Instruments category from Amazon.com. Both of the datasets have three relations, hence we treat them as multi-relation graphs. T-Social and T-Finance [40] are two large-scale datasets released recently. The T-Finance dataset aims to detect anomalous accounts in a transaction network where the nodes are annotated as anomaly if they are likely fraud, money laundering and online gambling. The nodes are accounts with 10-dimension features whereas the edges connecting them denote they have transaction records. The T-social dataset aims to detect human-annotated anomaly accounts in a social network. The node annotations and features are the same as T-Finance, whereas the edges connecting the nodes denote they maintain the friendship for more than 3 months.

## C LIMITED LABEL AND ANOMALIES

In the real-world case, the percentage of the anomaly is usually quite low, even less than 5% or even 1%; also human annotation is expensive which leads to limited label information. Hence we have conducted additional experiments to address this concern and provide insights into the performance in such scenarios in Table 5.

**Limited Label Information:** We experimented with reduced labeled data to 1%. Despite the reduced amount of labeled information, our proposed method still achieved good performance, demonstrating its ability to effectively leverage limited label information for accurate detection.

**Small Percentage of Abnormal Nodes:** We also examined the performance when the dataset contained a small percentage of abnormal nodes. In this scenario, our proposed method maintained a high level of f1 and auc in detecting the abnormal instances, even amidst the imbalanced class distribution.

**Small Percentage of Abnormal Nodes with limited label information:** In this case, the performance of the proposed method drops a little due to the inaccurate prediction of the NLD of the nodes.

