# OpenReview forum: "Graph Anomaly Detection with Bi-level Optimization"
_ACM.org/TheWebConf/2024/Conference — TheWebConf24_

### Official Review · Reviewer_wLyQ · 2023-11-16

**Novelty:** 4
**Technical Quality:** 5

**Review:**

The paper addresses challenges in Graph Anomaly Detection (GAD) using Graph Neural Networks (GNNs). Traditional GNNs often aggregate embeddings from all neighbors without considering their labels, hindering the detection of anomalies. Previous methods selectively aggregate neighbors, but this selection strategy is consistent for both normal and anomalous classes, limiting effectiveness. The study introduces the concept of "loss rivalry," observing that nodes with different classes yet similar neighbor label distributions tend to have opposing loss curves. The proposed solution, BioGNN, utilizes a Contextual Stochastic Block Model (CSBM) and NLD distance to segregate nodes based on classes and NLD at a lower level, while the upper level trains the anomaly detector using separation outcomes. Experimental results show that BioGNN outperforms existing methods and effectively mitigates the "loss rivalry" phenomenon.




**Strong Points:**

- Clarity and Accuracy in Descriptions: The paper is commended for its mostly clear descriptions with no typos. The methodology section could be improved with more intuition.

- Significance of Research Area: The paper addresses an important research area, Graph Anomaly Detection (GAD), which has applications in finance, healthcare, and security. This significance contributes to the relevance and potential impact of the research.

- Good Results: The paper reports very good results, surpassing state-of-the-art methods. This is a strong indicator of the effectiveness of the proposed solution, BioGNN, and underscores its potential practical utility.

- Analysis of NLD: The authors bring a nice perspective to the analysis of Neighbor Label Distributions (NLD) by incorporating the Contextual Stochastic Block Model (CSBM) from a graph generation standpoint.



**Weak Points:**

- Formula Intuition: Some of the formulas, notably eq6, lack accompanying intuition in their descriptions. Providing more context or explanation for key formulas can enhance the understanding of readers, especially those not deeply familiar with the specific mathematical formulations.


- Lemma 4.1 Clarification: The paper has a weak point related to Lemma 4.1, where the lack of a clear definition or proof may cause confusion. If it is presented more as an observation rather than a formal lemma, it should be explicitly stated as such. Clearing up this ambiguity will improve the overall credibility of the theoretical foundations.


- Time Complexity Analysis: The time complexity analysis is mentioned but lacks a formal proof. The statement about the constant C in the O() notation, along with its resemblance to a statement from another source ([40]), raises concerns about originality of the paper.


**I read the rebuttal and all my concerns were sufficiently addressed.**

**Questions:**

1. would it make sense to consider the NLD using a k-step neighborhood?

2. Why is the input graph complete?

3. Please elaborate on the running time complexity. What is C? Why is the constant mentioned in the O() notation?

**Reviewer Confidence:**

2: The reviewer is willing to defend the evaluation, but it is likely that the reviewer did not understand parts of the paper

**Scope:**

3: The work is somewhat relevant to the Web and to the track, and is of narrow interest to a sub-community

---

### Official Review · Reviewer_9KcF · 2023-11-17

**Novelty:** 4
**Technical Quality:** 4

**Review:**

In this paper, the authors address the heterophily issue in a graph anomaly detection task. They first observe a phenomenon (i.e., loss rivalry) that the nodes with different class labels yet similar neighbor label distributions tend to have opposing loss curves and explain this phenomenon theoretically. Then, the authors propose BioGNN that has two key components: (1) a mask generator and (2) two well-designed GNN encoders. Through the experiments using real-world datasets, they demonstrate the effectiveness of BioGNN. The strengths and weaknesses of this paper are as follows.



- Strengths.

S1. The authors observed the loss rivalry phenomenon

S2. The authors proposed BioGNN that addresses the heterophile and thus improves the performance of the graph anomaly detection task

S3. The authors demonstrated the effectiveness of BioGNN through the following experiments with real-world datasets



- Weakness.

W1. One key paper is missing, which is critical. The authors need to show the key difference between their work and this work and compare the performances.

-       [1] Y. Gao et al., “Addressing Heterophily in Graph Anomaly Detection: A Perspective of Graph Spectrum,” In Proc. ACM WWW, 2023.

W2. The notations used in the paper need to be defined more precisely and explained with clarity.

-       For example, on page 3, 'alpha' is described as the spectrum of a one-hot vector, while on page 5, it is defined as a hyperparameter.

-       The explanations of the meaning behind each encoder (theta, phi, large phi) are missing.

W3. It is necessary to conduct an in-depth analysis of each component.

-      The authors need to conduct an ablation test to demonstrate the effectiveness of each component and verify whether each component effectively addresses its intended purpose.

**Questions:**

Q1. Compared to the paper [1], what are the strengths and weaknesses of the proposed method?

**Reviewer Confidence:**

3: The reviewer is confident but not certain that the evaluation is correct

**Scope:**

3: The work is somewhat relevant to the Web and to the track, and is of narrow interest to a sub-community

---

### Official Review · Reviewer_hHiV · 2023-11-22

**Novelty:** 3
**Technical Quality:** 5

**Review:**

## Updates after Authors' Response

I appreciate the detailed responses from the authors, which have addressed the questions I raised in my original review. I am happy to adjust the score to take into account the additional discussions and efforts the authors put in their response.

With that being said, I feel that there are many substantial updates (e.g., adding additional baselines, discussions to other related works, important clarification/contextual discussions to the theorems, etc.) being pledged in this response (and the responses to other reviews). **While I appreciate the significant efforts the authors made to include these additional results/discussions in the rebuttal, I feel these changes are too much to be reviewed only as patches scattered in the responses; they should be reviewed comprehensively in the form of a complete paper revision. For this reason, I would not champion for the acceptance of the current version of the work before seeing a completed revision.**

---
## Summary

This paper focuses on improving the Graph Neural Network (GNN) performance on graph anomaly detection in the form of semi-supervised binary node classification. Specifically, the authors propose to (1) separate the nodes into two sets using a learnable separation function, and (2) employ two separate graph encoder (one with a low-pass filter and one with a high-pass filter) for each set of nodes. Experiment results show that the proposed approach demonstrates mostly marginal improvement to the existing approaches. The authors also provide theoretical analysis from the perspective of Neighborhood Label Distribution (NLD) which they claim to support the proposed approach.

## Strengths

1. Fig. 4 provides a nice high-level illustration of the proposed approach.
2. The experiments are bing conducted on a extensive set of baselines (even though missing a baseline that is more relevant to the proposed approach).

## Weakness

1. **For theoretical analysis, Proposition 3.1 seems to be similar to the results of Equation (4) in the prior work [33].** Specifically, [33] has given detailed derivation on the connections between the distance of Neighborhood Label Distribution (NLD) with the distance of their expected hidden representation for a pair of nodes from two different classes under the Contextual Stochastic Block Model (CSBM) model. The authors should cite this prior work in their analysis and address the similarities between their analysis and the analysis in [33].
2. **It is hard to understand how the proposed approach is motivated in the theoretical analysis.** There are some confusing discussions that aim to justify the proposed approach based on theoretical analysis in Line 344-347:

    > Proposition 3.2 indicates that capturing the difference in spectral label distribution is equivalent to measuring the similarity between NLDs. Furthermore, the proposition elucidates that different nodes with similar NLD retain rather different frequency components.
    >

    These two claims seem to be in conflict with each other: if the first sentence is true, how can different nodes with similar NLD can retain rather different frequency components? Also, the overall reasoning here seems to be based on the 2nd sentence here: because abnormalities and normal nodes cannot be distinguished from NLDs, they need to be distinguished from the spectral perspective. If that is the case, then what is the purpose of Proposition 3.1 here?

3. **For the proposed approach, another relevant work that the authors didn’t address or compare against in their work is GBK-GNN** [A1]: GBK-GNN also employs a gated selection mechanism that adapts to homophilous or heterophilous connections, despite that [A1] apply the gated mechanism on the edge level for message passing, while this work applies it on the node level for selecting graph encoder. Given both works are employing this bi-kernel design, I think the authors should address their differences in the related works and add GBK-GNN as an additional baseline in their experiments.
4. **Some technical details of the proposed approach are not clearly described:**
    a. For the choice of $g_1$ and $g_2$ in Equation (13), Appendix B.1 only shows the choice of $\alpha$ and $\beta$ which does not specify how they mapped to $g_1$ or $g_2$.
    b. Section 4.3 (Initialization of BioGNN) feels unclear in its current writing. I would suggest the authors to add an illustration to show the end-to-end process of parameter initialization.
    c. Figure 5, 6 and their discussions are hard to understand.

## References

[33] Yao Ma, Xiaorui Liu, Neil Shah, and Jiliang Tang. 2022. Is homophily a necessity for graph neural networks?. In ICLR.

[A1] Lun Du, Xiaozhou Shi, Qiang Fu, Xiaojun Ma, Hengyu Liu, Shi Han, and Dongmei Zhang. "Gbk-gnn: Gated bi-kernel graph neural networks for modeling both homophily and heterophily." In *Proceedings of the ACM Web Conference 2022*, pp. 1550-1558. 2022.

**Questions:**

I would like to see the authors address the questions raised in the weakness section of the review, especially for points (1)-(3).

**Reviewer Confidence:**

4: The reviewer is certain that the evaluation is correct and very familiar with the relevant literature

**Scope:**

4: The work is relevant to the Web and to the track, and is of broad interest to the community

---

### Official Review · Reviewer_qZ8H · 2023-11-23

**Novelty:** 4
**Technical Quality:** 4

**Review:**

This paper presents an innovative approach in Graph Neural Networks (GNNs) for Graph Anomaly Detection (GAD). To address the shortcomings of traditional GNNs in GAD, which often inaccurately detect anomalies due to improper aggregation of neighbor embeddings, the paper introduces the Bi-level optimization Graph Neural Network (BioGNN). This proposed method incorporates the Contextual Stochastic Block Model (CSBM) and introduces the Neighbor Label Distribution (NLD) distance metric. BioGNN uniquely tackles the "loss rivalry" issue, where nodes with different classes but similar NLDs exhibit opposing loss curves, affecting model convergence. The model operates on two levels: segregating nodes based on classes and NLD at the lower level, and training the anomaly detector at the upper level. Through extensive experiments, BioGNN demonstrates superior performance over existing methods, effectively enhancing anomaly detection accuracy in various scenarios, including financial fraud detection and identifying misinformation in social networks, marking a significant advancement in the field of GAD.

Strengths:
1. The motivation of the paper is well justified. The paper successfully identifies and mitigates the "loss rivalry" phenomenon, where nodes of different classes but with similar Neighbor Label Distributions (NLD) have opposing loss curves. This is a significant advancement in improving model convergence and accuracy.
2. The use of the Contextual Stochastic Block Model (CSBM) and the introduction of NLD distance provide a theoretical foundation for the proposed method.
3. The background is presented nicely and the paper is organized and easy to follow.
4. Comprehensive experimental results under different settings are provided to demonstrate the effectiveness of the model.

Weaknesses:
1. The proposed method is strongly dependent on the assumptions of the Contextual Stochastic Block Model (CSBM).
2. The model is highly dependent on the accurate prediction of the Neighborhood Label Distribution (NLD). When the proportion of anomalous nodes in the processed dataset is low or the labeling information is limited, the model performance may be slightly degraded due to the lack of accurate prediction of node NLDs.
3. In the real-world case, the percentage of the anomaly is pretty small, (e.g., less than 5% or even 1%). In the experiment, the percentage of the labeled nodes is more than 40% and the numbers of anomalies for most datasets are more than 10% (e.g., 10% for Amazon and 205 for YelpChi). Can the proposed method could still achieve good performance with limited label information as the proposed method only aggregates the message from labeled neighbors to avoid getting the information from anomalies? The proposed method seems to heavily rely on the good performance of the masking strategy and the quality of the mask seems to rely on a large amount of label information.  My major concern is whether the proposed method still performs well for these highly imbalanced datasets or with limited labeled nodes. It's highly recommended to conduct experiments on some datasets with less percentage of anomalies.

**Questions:**

Q1: The theoretical basis of the model relies strongly on the assumptions of the CSBM. What are some ways to reduce the reliance on these assumptions and thus enhance the applicability and robustness of the model under different graph generation processes?

Q2. The model performance degrades slightly when the percentage of anomalous nodes in the dataset is low. Are there ways to improve the prediction accuracy of NLD, especially in the case of unbalanced data or limited labeling information?

**Reviewer Confidence:**

4: The reviewer is certain that the evaluation is correct and very familiar with the relevant literature

**Scope:**

3: The work is somewhat relevant to the Web and to the track, and is of narrow interest to a sub-community

---

### Official Review · Reviewer_vFv3 · 2023-11-29

**Novelty:** 6
**Technical Quality:** 4

**Review:**

The paper proposes a method named BioGNN for graph-based anomaly detection. BioGNN adopts a bi-level optimization approach, where the first level optimizes an encoder for masking the graph to segregate nodes based on neighbor label distribution (NLD) distance, and the second level optimizes the classifier, as the problem is cast as semi-supervised node classification problem. The problem studied by the paper is important, and may draw wide interests in the community. The approach proposed by the paper is novel and interesting. Overall the writing is well polished, despite of a few issues, such as references of undefined notations in Section 3.1 and 3.2, a few broken sentences/grammar mistakes (e.g., Figure 2 caption, and some sentences in Section 6 ). Another strength of the paper is that it makes the code available, which is great for fast reproducibility. One suggestion here is to make a readily runnable script or demo, so that interested readers could easily verify certain experiments.

One of my major concerns is the design of the low level optimization, since it is not quite intuitive that the encoder theta would serve its purpose of assigning different masks to nodes with the different class but similar NLDs. In other words, the paper doesn't make enough justification on how the actual model design is connected with the theoretical intention. Actually, in another perspective besides the NLDs, the model design proposed by the paper looks quite related to the recent graph cleaning/de-noising (some also call it sanitation) line of works. The low level optimization serves the purpose of filtering nodes of undesired distribution in this sense. One related work is titled Graph Sanitation with Application to Node Classification, where bi-level optimization technique is also applied.

Another concern is about the experiment. It'd be great if the authors could share some insights in why the learning curves of high and low frequency shown in Figure 5 are so unsmooth. Sudden and huge jumps could be seen in around 50-th epoch.

For the experiments, it'd be great if the main results shown in table 3 could also contain standard deviation/statistical significance results on several runs, since the metrics reported on certain models are quite close.

One question about the Section 4.3 is that why it mentions that the input data a complete graph rather than a ego net? Is it because the datasets are complete graph or any other reasons? Please clarify this question.

**Questions:**

See above

**Ethics Review Description:**

no ethic issues

**Reviewer Confidence:**

4: The reviewer is certain that the evaluation is correct and very familiar with the relevant literature

**Scope:**

4: The work is relevant to the Web and to the track, and is of broad interest to the community

---

### Decision · Program_Chairs · 2024-01-22

**Decision:**

Accept

**Comment:**

The paper introduces the Bi-level optimization Graph Neural Network (BioGNN) for graph-based anomaly detection, addressing the limitations of traditional Graph Neural Networks (GNNs) in this area. By focusing on the "loss rivalry" phenomenon and employing a bi-level optimization approach, BioGNN demonstrates potential advancements in graph anomaly detection (GAD), a field with wide applications in finance, healthcare, and security.

 ## Strengths:
 1. Novel Approach and Theoretical Foundation: The introduction of BioGNN, which uses a bi-level optimization approach and incorporates the Contextual Stochastic Block Model (CSBM) and Neighbor Label Distribution (NLD) distance, is innovative and theoretically grounded. This approach addresses the critical issue of "loss rivalry," as highlighted by several reviewers.
 2. Experimental Validation: The experimental results, conducted on various real-world datasets, demonstrate the superior performance of BioGNN over existing methods, reinforcing the paper's contribution to the field.
 3. Clarity and Organization: The paper is generally well-organized and clearly written, with a detailed methodology section and comprehensive experiments.

 ## Weaknesses:
 1. Methodological Clarifications and Justifications: Reviewers expressed concerns regarding the clarity of certain methodological aspects, including the design of the low-level optimization and the connection between theoretical intentions and the model design. Reviewer 1 and 4 pointed out the need for more justification in model design and clearer definitions of notations.
 2. Dependence on CSBM Assumptions and NLD Predictions: Reviewer 2 noted that the model's effectiveness is strongly dependent on the assumptions of the CSBM and the accuracy of NLD predictions, which may limit its applicability in certain real-world scenarios.
 3. Lack of Comparison with Relevant Baselines: Several reviewers, including Reviewer 3 and 4, mentioned the absence of comparisons with relevant baselines, such as GBK-GNN and other related works, which are critical for establishing the method's novelty and effectiveness.
 4. Need for More Detailed Analysis: Reviewers suggested conducting in-depth analysis, such as ablation tests, to assess the effectiveness of individual components of BioGNN. Reviewer 4 and 5 emphasized the need for clearer explanations of key formulas and the time complexity analysis.

 The authors' response indeed does a good job to address some of the above concerns, which makes several reviewers increase their evaluation after reading the response. However, some reviewers also mentioned that fully addressing the concerns may be too much and need another round of review, which I share some similar spirit. Overall, I give a borderline recommendation. The technical idea of this paper is good. Even if the paper gets rejected this time, after proper revision, this will become a strong paper for the future round of review.